# Straightforward Enzymatic Methacrylation of Poly(Glycerol Adipate) for Potential Applications as UV Curing Systems

**DOI:** 10.3390/polym15143050

**Published:** 2023-07-14

**Authors:** Víctor Hevilla, Águeda Sonseca, Marta Fernández-García

**Affiliations:** 1Instituto de Ciencia y Tecnología de Polímeros (ICTP-CSIC), C/Juan de la Cierva, 3, 28006 Madrid, Spain; v.hevilla@ictp.csic.es; 2Interdisciplinary Platform for “Sustainable Plastics towards a Circular Economy” (SUSPLAST-CSIC), 28006 Madrid, Spain; 3Instituto de Tecnología de Materiales, Universitat Politècnica de València, Camino de Vera, s/n, 46022 Valencia, Spain

**Keywords:** poly(glycerol adipate), enzymatic synthesis, functionalization, steps, UV-curing

## Abstract

Enzymatic one-pot synthesis procedures in a one-step and two-step monomers addition were developed to obtain poly(glycerol adipate) macromers with methacrylate end-functional groups under the presence of 1 and 3 wt% of *Candida antarctica* lipase B (CALB). Glycerol, divinyl adipate, and vinyl methacrylate were enzymatically reacted (vinyl methacrylate was either present from the beginning in the monomers solution or slowly dropped after 6 h of reaction) in tetrahydrofuran (THF) at 40 °C over 48 h. Macromers with a methacrylate end groups fraction of ≈52% in a simple one-pot one-step procedure were obtained with molecular weights (M_n_) of ≈7500–7900 g/mol. The obtained products under the one-pot one-step and two steps synthesis procedures carried out using 1 and 3 wt% of a CALB enzymatic catalyst were profusely characterized by NMR (^1^H and ^13^C), MALDI-TOF MS, and SEC. The methacrylate functional macromers obtained with the different procedures and 1 wt% of CALB were combined with an Irgacure^®^ 369 initiator to undergo homopolymerization under UV irradiation for 10 and 30 min, in order to test their potential to obtain amorphous networks within minutes with similar properties to those typically obtained by complex acrylation/methacrylation procedures, which need multiple purification steps and harsh reagents such as acyl chlorides. To the best of our knowledge, this is the first time that it has been demonstrated that the obtention of methacrylate-functional predominantly linear macromers based on poly(glycerol adipate) is able to be UV crosslinked in a simple one-step procedure.

## 1. Introduction

Nowadays, there is an increasing interest in the synthesis of polyester macromers based on a variety of monomers with different functional groups. More advantageous is to control the polymer end groups’ nature, since terminal functionalized polymers/macromers are very useful; for instance, they are used as blocks for new copolymers, comb polymers, and polymer networks for surface modification applications [1,2,3,4,5,6]. Conventional strategies to afford polymers with abundant functionalities are frequently based on complex processes that often involve multi-step synthesis procedures with intermediate purification steps that are tedious and time-consuming. Thus, more environmentally friendly procedures and a faster and higher implementation simplicity are desired. This demands efficient one-pot synthesis approaches to obtain the desired product in a single reaction, which, at the same time, will permit the control of the polymer structure [7,8,9,10,11,12]. To this end, the excellent chemo, regio-, and stereoselectivity of enzymes are of interest in endowing polyesters with specific end-functionalities in one-pot processes [13]. However, mostly, the aim of polyester synthesis using enzymes as catalysts has been strongly focused on obtaining high-molecular-weight polymers and its success in scaling them up, with the benefit of mild reaction conditions, more than obtaining specific functionalities in a single-step procedure [14,15,16,17].

Polyols such as glycerol are interesting due to their high hydroxyl functionality that provides an enhanced hydrophilicity, reduced toxicity, and high susceptibility to biodegradation [18,19]. Additionally, pendant hydroxyl groups can permit further functionalization, drug encapsulation, and the conjugation of active molecules [20,21,22]. Typically, to achieve structural homogeneity in glycerol-based polymers, a high linearity must be ensured. Towards this aim, a common practice is to selectively protect/block secondary hydroxyl groups [23,24], although the use of enzymes as selective catalysts with a preference for a primary hydroxyl reaction is a promising but less exploited alternative. In previous works, we have demonstrated the possibility of achieving linear poly(glycerol adipate) (PGA)-based macromers enzymatically synthesized from divinyl adipate and glycerol with controlled end group functionality by the means of reagents ratios, synthesis conditions, and amounts of catalyst [25].

Nowadays, 3D printing, such as stereolithography, has aroused great interest due to the rapid and easy preparation of polymer systems based on biodegradable polymers, which mitigate the concern about the use of non-renewable resources in this field [26]. These biobased polymeric systems have potential and versatility in biomedical applications [27,28], mainly because they present mechanical properties similar to soft tissues, and biocompatibility [29,30,31,32,33]. Therefore, it would be of greater interest to obtain thermoset polyesters from these macromers. For polyester macromers to be rendered photocurable within minutes under the presence of a photoinitiator and the appropriate wavelength light, their functionalization with acrylate/methacrylate moieties, typically incorporated from the reaction of acyl chlorides in the presence of triethylamine, is one of the preferred procedures [34,35]. These methods imply harsh reagents that need to be neutralized during the process, resulting in a large number of undesirable side products that are difficult to remove; thus, multiple purification/washing steps are needed. Moreover, due to the high reactivity of the acyl chlorides, the structural homogeneity of the macromers is compromised unless the protection of secondary hydroxyl groups is performed. Our group also performed, for the first time, enzymatic functionalization with methacrylate moieties in the independent synthesis step of a previously enzymatically synthesized and purified PGA macromer bearing a majority of hydroxyl end groups [36].

Importantly, the amount of substitution degree of the hydroxyl end groups with methacrylate moieties allowed for film formation during the photocrosslinking of this linear macromer. However, this synthesis procedure was carried out in two well-differentiated syntheses—firstly, a PGA macromer was enzymatically synthesized from divinyl adipate and glycerol monomers into THF for 24 h. The product was then subsequently isolated from the reaction media and purified to obtain the desired macromer; secondly, the obtained PGA was subsequently redissolved in THF for its enzymatic functionalization, in an independent synthesis, under the presence of vinyl methacrylate for 48 h. Although the obtained macromers presented the desired characteristics, the synthesis protocols were time and resource demanding. These synthesis protocols could be improved if these macromers could be obtained in a one-step reaction. This improvement would mean the use of less catalyst, solvent, and time.

Therefore, our aim in this work was to study and methacrylate functionalized linear poly(glycerol adipate) macromers using one-pot step lipase catalyzed polycondensation/transesterification and functionalization. This synthetic protocol is not only a more ecological and simpler alternative to the one used to obtain the previous crosslinkable polymer, but also produced a greater modification of the macromers. Additionally, the ability of the obtained macromers under ultraviolet light radiation to undergo further crosslinking and film formation via initiated free-radical polymerization was demonstrated.

## 2. Experimental

### 2.1. Materials

Glycerol (GLY 99%, Sigma-Aldrich, St. Quentin Fallavier, France), divinyl adipate (DVA, 96.0%, TCI Europe, Paris, France), vinyl methacrylate (VMA, TCI Chemicals, Paris, France, 98.0%), acetone (99.5%, Scharlau, Sentmenat, Barcelona, Spain), diethyl ether (99.8%, Honeywell, Charlotte, NC, USA), 2-benzyl-2-dimethylamino-1-(4-morpholinophenyl)-1-butanone (Irgacure^®^ 369, Ciba Specialty Chemicals, Basel, Switzerland), deuterated acetone (99.8%, Eurisotop, Andover, MA, USA), N,N-dimethyl formamide (DMF, Scharlau), and lithium bromide (LiBr, Sigma Aldrich, >99.9%) were used as received. Anhydrous tetrahydrofuran (THF) was obtained by passing analytical grade (Scharlab) over a solvent purification system (Innovative Technology, Inc., Newburyport, MA, USA). Lipase B from *Candida antarctica*, immobilized on microporous (CALB (Novozyme 435) 5000 U/g, Sigma-Aldrich), was dried for 24 h before its use.

### 2.2. Synthesis of Methacrylated Poly(glycerol Adipate) Macromers

#### 2.2.1. One-Pot Two-Step Enzymatic Synthesis of Methacrylated Poly(glycerol Adipate)

For the two-step one-pot synthesis, DVA (0.760 g, 3.9 mmol) and GLY (0.389 g, 4.22 mmol) were dissolved in anhydrous THF (730 mg/mL relative to the total monomers weight) in a round-bottom flask. CALB was added in different amounts (1–3 wt% of the total monomers weight) and the mixture was allowed to react for 6 h under argon and magnetic stirring at 40 °C. After that time, VMA (0.258 g, 2.30 mmol) was slowly dropped over 30 min into the reaction media and allowed to react (Figure 1). After 48 h of total reaction time, THF was added to dilute the products and the reaction medium was poured into a glass funnel with a filter plate to facilitate the separation of enzymes. After filtering, the solutions were concentrated and dried into a rotary evaporator, subsequently dissolved in acetone, and precipitated into cold diethyl ether. Precipitates were recovered using decantation and dried under vacuum at room temperature until a constant weight was obtained, so no residual solvents were detected during characterization. NMR and MALDI-TOF analyses were carried out over the obtained products.

#### 2.2.2. One-Pot One-Step Enzymatic Synthesis of Methacrylated Poly(glycerol Adipate)

For the one-step one-pot synthesis, DVA (0.760 g, 3.83 mmol), GLY (0.389 g, 4.22 mmol), and VMA (0.258 g, 2.30 mmol), (1.0:1.1:0.6) were dissolved in anhydrous THF (730 mg/mL relative to the total monomers weight) in a round-bottom flask. CALB was added in different amounts (1–3 wt% of the total monomers weight) and the reaction was left under argon and magnetic stirring for 48 h at 40 °C (Figure 1). After the reaction time, the products were recovered as those in the previously described one-pot two-step enzymatic synthesis description were. NMR and MALDI-TOF analyses were carried out over the obtained products.

### 2.3. UV Curing and Film

To prepare photo-crosslinkable systems, PGA-Met obtained macromers were dissolved in acetone (263 mg/mL), and 3 wt% of a photoinitiator (Irgacure^®^ 369), concerning the total mass of the PGA-Met, was added to the solutions (Figure 2). The obtained mixtures were protected from light and stirred for 5 min at room temperature. The Crosslinked samples were prepared on glass plates (ø = 12 mm), adding 10 μL of mixture and drying them for 10 min in a fume hood protected from light at room temperature, allowing the solvent to evaporate. Subsequently, they were exposed to a UV lamp using a UVPTM CL–1000 short-wave photo-crosslinker (λ = 313 nm) for 10 and 30 min. The photo-crosslinked samples were named as n-mPGA-Metx, n being the amount of CALB, m the number of steps, and x the time of UV irradiation during crosslinking. Thus, 1-2PGA-Met10 indicates the crosslinked sample from a PGA-Met macromer obtained with 1 wt% of CALB in a one-pot two-step synthesis procedure and crosslinked by exposure to UV radiation for 10 min.

### 2.4. Characterization

#### 2.4.1. Nuclear Magnetic Resonance, ^1^H and ^13^C NMR analyses

^1^H (128 scans, 1 s relaxation delay) and ^13^C NMR (5120 scans, 1 s relaxation delay) spectra were recorded on a TM Bruker DPX 400 (400 MHz) spectrometer in deuterated acetone (≃40 mg/mL for ^1^H NMR and ≃80 mg/mL for ^13^C NMR), containing tetramethylsilane (TMS) as an internal standard. The amount of 1,2,3-substituted (branching degree) and 1,2-substituted units were calculated from the ^1^H NMR, following Equations (1) and (2):(1)1,2,3−substituted units=Ie″ne″Iana·100(2)1,2−substituted units=Ie′ne′Iana·100
where I is the intensity and n is the number of protons.

The degree of branching was calculated using the following Equation (3):(3)Degree of Branching (DB)=2D/(2D+L)·100

#### 2.4.2. Matrix-Assisted Laser Desorption/Ionization-Time-of-Flight Mass Spectrometry (MALDI-TOF-MS)

A matrix-assisted laser desorption/ionization-time-of-flight mass spectrometry (MALDI-TOF-MS) analysis of the synthesized macromers was performed using a Voyager-DE PRO time-of-flight mass spectrometer (Applied Biosystems, Foster City, CA, USA) equipped with a nitrogen laser emitting at λ = 337 nm. All the spectra were acquired in positive ion mode and the delayed extraction and instrumental settings were tuned to the parameters that optimized the signal intensity and resolution. The samples were dissolved in acetone and 1 µL of matrix (2,5-dihydroxybenzoic acid (DHB)) and 1 µL of cesium chloride were mixed with 1 µL of each dissolved sample (1 mg/mL).

#### 2.4.3. Size Exclusion Chromatography (SEC)

The obtained macromers were analyzed using Size Exclusion Chromatography (SEC) to estimate the number-average molecular weight (M_n_), weight-average molecular weight (M_w_), and polydispersity index (Đ). DMF stabilized with 0.1 M LiBr was used as an eluent at a temperature of 50 °C and 1 mL min^−1^ flow rate. A Waters Division Millipore system equipped with a Waters 2414 refractive index detector and Styragel packed columns (HR2, HR3, and HR4, Waters Division Millipore, Billerica, MA, USA) was used. The calibration curve was obtained using poly(methyl methacrylate) standards (Polymer Laboratories LTD) ranging from 1.4 × 10^6^ to 5.5 × 10^2^ Da. The samples were dissolved in DMF and filtered through a 0.45 µm Teflon filter before the analysis.

#### 2.4.4. Differential Scanning Calorimetry (DSC)

The thermal behavior of the photocrosslinked films was studied with a TA Instruments (New Castle, DE, USA) DSC Q2000 differential scanning calorimeter, equipped with a refrigerated cooling system (RCS) under an N_2_ atmosphere. The measurements were carried out in a three-step procedure of heating–cooling–heating over the temperature range from −70 to 180 °C. After being equilibrated at −70 °C, the samples were heated to 150 °C at 10 °C/min, then rapidly cooled to −70 °C, and heated again to 180 °C at the same rate. The glass transition temperatures (T_g_) were estimated from the first and second heating runs and taken as the temperature at the midpoint of the calorific capacity change.

#### 2.4.5. Fourier-Transform Infrared Spectroscopy with Attenuated Total Reflectance (FTIR-ATR)

The crosslinking process was confirmed by analyzing the samples in a Perkin Elmer Spectrum Two instrument equipped with an attenuated total reflection module (ATR). A background spectrum was acquired before every sample and all the spectra were recorded between the 400 and 4000 cm^−1^ spectral range with a 4 cm^−1^ resolution.

## 3. Results and Discussion

### 3.1. Molecular Weight and Structural Characterization

#### 3.1.1. ^1^H NMR, ^13^C NMR and SEC

The one-pot procedure was carried out by reacting GLY, DVA, and VMA dissolved in anhydrous THF with CALB at 40 °C under N_2_ and continuous stirring. For the two-step one-pot procedure, the VMA monomer was added to the mixture after 6 h of the DVA and GLY reaction (see Figure 1).

Figure 1A,B show the ^1^H NMR assignments and spectra of the products, respectively. The assignments and calculations were made based on our previous studies of enzymatically synthesized PGA with different reagent ratios, catalyst amounts, and others [25]. CH_2_ adipic protons (a, b) appeared in the spectral range between 1.6 and 2.5 ppm, while all the protons related to glyceride repeating units were found between 3.5 and 5.3 ppm; (c, d) and (e) protons corresponded to a 1,3-substituted linear glycerol unit; (h, i), (f, g), and (e’) protons correspond to a 1,2-substituted linear glycerol unit; and (h’, i′), and (e″) corresponded to a 1,2,3-substituted dendritic glycerol unit. The last peak (dendritic, e″) at 5.3 ppm due to the 1,2,3-substituted glycerides was produced due to the lack of regioselectivity of the enzyme during the reaction. Glycerol terminal units gave multiplets in the region of 3.4-3.7 ppm, (m, n), (j, k), and (l) protons corresponded to terminal glycerol that was 1 or 3 substituted, while (m′, n′) and (l′) protons corresponded to terminal glycerol that was 2 substituted. Resonances at 1.8, 5.5, and 6 ppm were attributed to the (p), (o′), and (o) protons of the vinyl component from the methacrylate group, respectively.

Figure 2A,B show the ^13^C NMR assignments and a representative spectrum of the product obtained in the one-pot one-step synthesis with 3 wt% of CALB. The ^13^C NMR of the macromers obtained in the one-pot one-step synthesis with 1 wt% of CALB and two-step synthesis with 1 and 3 wt% of CALB are shown in Appendix A in Figure A1 and Figure A2, respectively.

^1^H NMR and ^13^C NMR confirmed that the methacrylate end groups were bounded to the poly(glycerol adipate) polymer when 3 wt% of CALB was present, either for the one-step or two-steps one-pot syntheses. For smaller amounts of CALB (1 wt%), methacrylate end groups were only found in the one-pot synthesis strategy. When VMA was added after a 6 h reaction of DVA and GLY, resonances from the methacrylate group protons/carbons were absent, while vinyl protons/carbons from the DVA appeared. This fact can be indicative of the VMA reacting at the latest stages of the synthesis, when most of the DVA was incorporated into the polymer chains. This could be due to the slower reaction rate of VMA, which acts as an end-capper terminating the polymeric chains’ growth, as well as to the presence of the higher amount of vinyl groups suffering from potential hydrolysis, which could slow down the kinetics hydrolysis of the DVA, making it able to participate for longer in the chain growth and not lose the vinyl end groups.

As we previously reported [25], the ^13^C NMR spectra of PGAs facilitate the verification of the presence of terminal glycerol units, as well as the hydrolysis of double bonds from terminal divinyl adipate, which are not clearly revealed in ^1^H NMR spectra. Figure 2A,B display the carbon peak assignments, based on previous studies, as well as expanded regions of 61–77 ppm, 125–138 ppm, and 173–174 ppm [37,38,39,40]. Importantly, the spectrum for the one-pot two-step protocol with the highest amount of CALB (3 wt%) shows a clear, intense signal centered at 174.6 ppm, which corresponds to an acid carbonyl carbon due to the DVA end-group hydrolysis. Therefore, the low M_n_ and M_w_ achieved in this polymer could be strongly related to the hydrolysis of the vinyl groups at the early synthesis stages, which was somehow prevented when VMA was present as a starting monomer (one-pot synthesis procedure). In such a situation, an increase in the availability of vinyl groups from different species in the reaction mixture would permit DVA to contribute to the polymer chain growth to a higher extent. For a lower amount of CALB (1 wt%) in the two-step protocol, the ^13^C NMR spectrum confirms the slower reaction rate of the VMA. In agreement with the ^1^H NMR results, the peaks at 142 ppm and 97 ppm were due to the reacted DVA vinyl end groups being preserved, while the peaks centered at 18.4 ppm (Q), 125.8 ppm (O), and 137.3 ppm (P) were due to methacrylate end groups being absent.

In line with this, the data analysis of Table 1 shows that there is a significant difference among the M_n_ and M_w_ achieved in the one-pot one-step and two-step synthesis procedures, resulting in the highest values when 3 wt% of CALB was added to the reaction mixture. When VMA was present as a starting monomer, a higher M_n_ and M_w_ were reached (M_n_; 7450–7900 g/mol and M_w_; 12,100–13,600 g/mol for 1 and 3 wt%, respectively), while adding this monomer when the DVA and GLY were already reacted for 6 h considerably diminished the M_n_ and M_w_ values, being more noticeable for the lowest amount of CALB (M_n_; 1150 g/mol and M_w_; 1220 g/mol). This confirms that a higher presence of potential hydrolyzable groups from the beginning of the synthesis helped to preserve the reactive ends of the DVA, and therefore to achieve higher molecular weights. In addition to this, the Ð_SEC_ values indicate the formation of branched polymers by the incorporation of DVA at the beginning of the reaction.

The relative percentages of TG (1-substituted), TG2 (2-substituted), L1-3 (1,3-substituted), L1-2 (1,2-substituted), and D (1,2,3-substituted), and the end group relative abundance in terms of the % of glycerol, divinyl, and methacrylate end groups, as well as the regioselectivity, degree of branching, and degree of methacrylation, are given in Table 2. The results compiled there show that all the synthesized polymers were mainly linear, as DB was retained at really low values, which is consistent with our previous report on enzymatically synthesized PGA polyesters with a low excess of vinyl groups [25]. Thus, the regioselectivity of the enzyme was retained and methacrylate moieties were preferentially located at the end positions of the polymeric chains. Interestingly, the higher the amount of VMA end groups, the lower the 2-substituted glycerol end groups; thus, it seems that most of the VMA groups were incorporated by reacting with primary the hydroxyl groups and that the regioselectivity lack was mainly due to the late incorporation into the secondary hydroxyl groups from glycerol in the polymer chains.

#### 3.1.2. MALDI-TOF-MS

A representative MALDI-TOF spectrum from a PGA-Met obtained in the one-pot synthesis with 3 wt% of CALB is shown in Figure 3. The analysis showed several distributions corresponding to different degrees of polymerization of the PGA terminated with various groups (see Figure 3, Figure A3, Figure A4, Figure A5 and Figure A6 and Table A2): n_gv_ stands for HO-(PGA)-Vinyl (202n + 44), n_vv_ stands for Vinyl-(PGA)-Vinyl (202n + 198), n_gg_ stands for HO-(PGA)-OH (202n + 92), n_gm_ stands for HO-(PGA)-Met (202n + 86), n_mm_ stands for Met-(PGA)-Met (202n + 170), and n_mv_ stands for Met-(PGA)-Vinyl (202n + 156) structures. Due to the hydrolysis reactions of the vinyl groups during the growth of the polymers, free acid ends were also present, where: n_ga_ stands for HO-(PGA)-Acid (202n + 18), n_aa_ stands for Acid-(PGA)-Acid (202n + 146), n_va_ stands for Vinyl-(PGA)-Acid structures (202n + 172), and n_ma_ stands for Met-(PGA)-Acid (202n + 158) structures.

Wherein, n is the number of peaks per polymer chain. The repeating mass difference between any given series of peaks is 202 Da, which is the molecular weight for a linear hydroxyl-substituted polyester from GLY and DVA.

Figure 4 and Table 3 show the end group composition of the polyesters obtained in the one-pot one-step and two-step synthesis procedures under the presence of 1 and 3 wt% of an enzymatic catalyst. Interestingly, the one-step synthesis procedure allowed for the highest proportion of Met-(PGA)-Met (% n_mm_) end-capped-type chains (~52–55%), without a marked influence on the enzyme concentration. When polymers were obtained in the one-pot two-step synthesis procedure, the number of Met-(PGA)-Met species significantly decreased, being higher when CALB was present in a higher amount (3 wt%), in accordance with the ^1^H NMR results.

As shown in Table 3, importantly, the terminal carboxylic groups that were not evidenced in the ^1^H NMR were evidenced by MALDI-TOF-MS, confirming the ^13^C NMR results. The quantitative determination of the acid end groups using MALDI-TOF shows that the maximum acid chain ends were obtained for the highest amount of catalyst in the one-pot two-step synthesis procedure. For 3 wt% of CALB and the one-pot two-step synthesis procedure, it seems that the highest amount of catalyst hydrolysed the DVA end groups, increasing the amount of acid terminal chains.

### 3.2. Thermal Properties of UV Crosslinked Films

Having in mind that the polymers synthesized with 1 wt% of CALB presented a lower hydrolysis and had a similar percentage to those obtained with 3 wt% of CALB, the evaluation of their applicability as photo-crosslinking systems was performed with polymers synthesized with a lower amount of lipase and considered as less costly. The thermal properties of the PGA-Met macromers obtained in the one-pot one-step and two-step synthesis procedures after UV irradiation for 10 and 30 min (see Figure 2), 1-1PGA-Met10, 1-1PGA-Met30, 1-2PGA-Met10, and 1-2PGA-Met30, respectively, were analyzed using DSC and the resulting T_g_s are collected in Table 4.

The crosslinking of selected macromers resulted in amorphous networks, as only glass transitions were visible in DSC, with no evident melting or crystallization. Importantly, the T_g_ in the second heating run increased at around 10 °C for all the samples, which was indicative of the presence of some uncured fraction that was suppressed after the first heating run. With increasing the amount of VMA end groups (see Table 3), as well as the molecular weight of the macromers (see Table 1), the T_g_ value also increased, which was indicative of the higher crosslinking degree obtained for the macromer synthesized in the one-pot one-step procedure with 1 wt% of CALB. Additionally, no noticeable changes in the transitions were observed after 30 min of UV irradiation in comparison to the values obtained at 10 min. This indicates that curing longer times were not able to crosslink the remaining fractions; therefore, the shorter time was enough to obtain the UV curing of the samples. Moreover, the samples synthesized in two steps did not produce consistent films, since their molecular weights were small and percentage of VMA end groups was lower.

Figure 5 shows the FTIR spectra of the systems before and after exposure to UV light at 10 and 30 min. It is easily appreciable that the band at 1660 cm^−1^, corresponding to the stretching vibration of the methacrylate double bond, disappeared, while a new band at 1595 cm^−1^, corresponding to -CH_2_-CH_2_-, appeared. It is also observed that still there were double bonds available due to the elevated percentage of functionalization, which was not able to fully cure itself. However, it could be a promising candidate material for stereolithography.

## 4. Conclusions

Our main goal was to create enzymatically synthesized methacrylated end functional poly(glycerol adipate) linear macromers in a one-pot synthesis procedure, which were able to undergo film formation under UV light irradiation when a photoinitiator was added. Polyester macromers with a high methacrylate content were obtained in a simple one-pot one-step enzymatic synthesis (≈52% of methacrylate end groups moieties in comparison to ≈25% for two-step enzymatic synthesis, as calculated using MALDI-TOF-MS). The ^13^C NMR analysis confirmed the presence of acid end groups due to the DVA hydrolysis, which was quantified using MALDI-TOF-MS to be in the range of ≈20% for the one-pot one-step-obtained macromers. The highest amount of methacrylate functionality of one-pot one-step synthesized macromers with 1 wt% of CALB allowed for the formation of higher crosslinking networks. Thus, the synthesized macromers included polyesters with different combinations of methacrylate, vinyl, hydroxyl, and carboxyl acid end groups, which could be shifted towards obtaining preferentially methacrylate end-capped species in a higher concentration by using a one-pot one-step synthesis method. This easy and clean synthesis approach towards methacrylate functional polyesters that is useful for photo-crosslinking processes has not been previously reported and constitutes, to date, the most efficient procedure, opening up a promising strategy for industrially applicable new biobased materials for 3D printing.

## Data Availability

Not applicable.

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
