# Peer review of "Straightforward Enzymatic Methacrylation of Poly(Glycerol Adipate) for Potential Applications as UV Curing Systems"

_polymers, 2023, doi:10.3390/polym15143050_

Round 1

Reviewer 1 Report

1. In the introduction to the article, the problem of using new polymers analogous to those obtained in the work is not discussed at all. Although there is a phrase in the conclusion that such polymers have the prospect of an industrially applicable new bio-based material for 3D printing. The results of studies on the practical application of analogues of new materials should be added in the introduction.

2. In addition, the introduction should clearly formulate the goals and objectives of the study, the approaches used to solve them.

3. Anhydrous THF is used in the synthesis of new polymeric materials. In the experimental part, it should be indicated which commercial THF was used in the work, how was it prepared for use in the synthesis? How was anhydrous THF obtained? There are several options.

4. The polymer after synthesis was separated by filtration. There are various types of filtering. Specify which type of filtering was used.

5. After isolation, the polymer was dried in vacuo. At what temperature? The drying temperature must be specified.

6. In the discussion of the results, the molecular weight characteristics of the new polymers are discussed. However, the differences in the Mw/Mn polydispersity coefficients are not discussed at all. Why are the polydispersity coefficients Mw/Mn higher in two-step synthesis? Should be discussed and explained.

7. The topic of the article is interesting and relevant, and in the list of references there are references to publications mainly until 2017. New works in the area under consideration should be added to the list of references.

Author Response

First of all, we would like to thank the reviewer for his/her exhaustive reading. 

  1. In the introduction to the article, the problem of using new polymers analogous to those obtained in the work is not discussed at all. Although there is a phrase in the conclusion that such polymers have the prospect of an industrially applicable new bio-based material for 3D printing. The results of studies on the practical application of analogues of new materials should be added in the introduction.

We have introduced a new paragraph in the Introduction.

  1. In addition, the introduction should clearly formulate the goals and objectives of the study, the approaches used to solve them.

We have modified the text to clarify our work.

  1. Anhydrous THF is used in the synthesis of new polymeric materials. In the experimental part, it should be indicated which commercial THF was used in the work, how was it prepared for use in the synthesis? How was anhydrous THF obtained? There are several options.

The preparation was added in the Materials Section.

  1. The polymer after synthesis was separated by filtration. There are various types of filtering. Specify which type of filtering was used.

Thanks for this comment. There was no real filtration but decantation. This was corrected in the manuscript.

  1. After isolation, the polymer was dried in vacuo. At what temperature? The drying temperature must be specified.

It was done at room temperature.

  1. In the discussion of the results, the molecular weight characteristics of the new polymers are discussed. However, the differences in the Mw/Mn polydispersity coefficients are not discussed at all. Why are the polydispersity coefficients Mw/Mn higher in two-step synthesis? Should be discussed and explained.

A new paragraph about the polydispersity indexes was added to the manuscript.

  1. The topic of the article is interesting and relevant, and in the list of references there are references to publications mainly until 2017. New works in the area under consideration should be added to the list of references.

We have introduced new and recent references. 

Reviewer 2 Report

Dear authors

Please find enclosed a review report

Reviewer

Author Response

First at all, we would like to thank the reviewer for his/her exhaustive reading.

1. The article is good, interesting and contains a great deal of originality. Nevertheless, there is a need for additions and clarification, as follows. I suggest that the title be changed: Suggestion: Straightforward enzymatic methacrylation of poly(glycerol adipate) as UV-curing systems

The title has been changed as suggested by reviewer

2. Abstract: The full name of the abbreviations that appear for the first time, for example CALB.

It was corrected.

3. Also, Irgacure 369 initiator is necessary to define by chemical composition.

We have the IUPAC name of commercial Irgacure®, 2-benzyl-2-dimethylamino-1-(4-morpholinophenyl)-1-butanone.

4. In the abstract is not mentioned variation of UV curing. I suggest mentioning this variation as well.

As suggested by the reviewer we have included the time of curing in the abstract.

5. P 3, L100, drying protocol and conditions should be described.

It has been described.

6. The figures are partly unclear (NMR), the resolution should be increased.

We have tried to improve the images.

7. Table 1 is not explained.

We have included an explanation of Table 1.

8. Edit the font size in Table 2, 3.

Done.

9. Explain the choice of spectrum from 2500 to 3200 m/z for the data in Figure 3.

Thanks for this comment. We made a mistake; the range is from 1620 a 2260 m/z. We selected this range of the spectrum because it is an area where the different final groups present in the PGA macromer can be observed for different repeating units (n), which is marked with ∆.

10. In the conclusion it was mentioned that the novel polymer is suitable for 3D printing! I think it is necessary to combine this application with existing polymers for these purposes in the introduction part.

Most of 3D printing systems in stereolithography are based on acrylic monomers; therefore, from oil-based polymers and not from biobased ones. We have introduced a paragraph about this.